



# A multi-decadal time series of upper stratospheric temperature profiles from Odin-OSIRIS limb scattered spectra

Daniel Zawada[1], Kimberlee Dubé[1], Taran Warnock[1], Adam Bourassa[1], Susann Tegtmeier[1], and Douglas Degenstein[1]

[1]Institute of Space and Atmospheric Studies, University of Saskatchewan, Saskatoon, SK, Canada

**Correspondence:** Daniel Zawada (daniel.zawada@usask.ca)

**Abstract.** A new upper stratospheric (35-60 km) temperature data product has been produced using Optical Spectrograph and InfraRed Imager System (OSIRIS) limb scattered spectra that now spans over 22 years. Temperature is calculated by first estimating the Rayleigh scatter signal, and then integrating hydrostatic balance combined with the ideal gas law. Uncertainties are estimated to be 1–5 K, with a vertical resolution of 3–4 km. Correlative comparisons with the Atmospheric Chemistry

Experiment Fourier Transform Spectrometer (ACE-FTS) and the Microwave Limb Sounder (MLS) are consistent with these uncertainty estimates, and generally have no regions of statistically significant drift. The data product has been publicly released as part of the nominal OSIRIS v7.3 processing.

## 1 Introduction

Stratospheric cooling over the past several decades is a key sign of anthropogenic climate change (e.g., Gulev et al., 2021). In

order to monitor temperature variations on climate time scales it is necessary to have an observational dataset that extends for multiple decades: it has thus far been difficult to quantify the magnitude of stratospheric cooling due to a deficit of long-term and vertically resolved observational datasets, particularly in the upper stratosphere where observations from Global Navigation System (GNSS) Radio Occultation (RO) measurements are highly uncertain (Steiner et al., 2020a). Temperature trend studies in the middle and upper stratosphere have largely relied upon merged datasets with limited vertical resolution (e.g., Randel

et al., 2016; Steiner et al., 2020b; Maycock et al., 2018; Randel et al., 2017). Global trends from these studies for the ozone recovery period (post ∼1998) range from -0.19 K/decade (Randel et al., 2016) to -0.5 K/decade (Steiner et al., 2020b) near 40 km, and from -0.28 K/decade (Randel et al., 2016) to -0.6 K/decade (Steiner et al., 2020b) near 45 km. This relatively large uncertainty in upper stratospheric cooling limits our ability to understand the response of the stratosphere to anthropogenic climate change, and motivates the development of a new stratospheric temperature product.

Several studies have demonstrated the ability to retrieve stratospheric and mesospheric temperature profiles from limb scatter measurements. Sheese et al. (2012) used spectra from the Optical Spectrograph and Imager System (OSIRIS, Llewellyn et al., 2004) to determine temperatures in the range 55–80 km, and Hauchecorne et al. (2019) used Global Ozone Monitoring by Occultation of Stars (GOMOS, Kyrölä et al., 2004) daytime limb observations to retrieve temperature in the 35–85 km range. While these methods differ in algorithmic details, they share a common core procedure of using isolated wavelengths in the



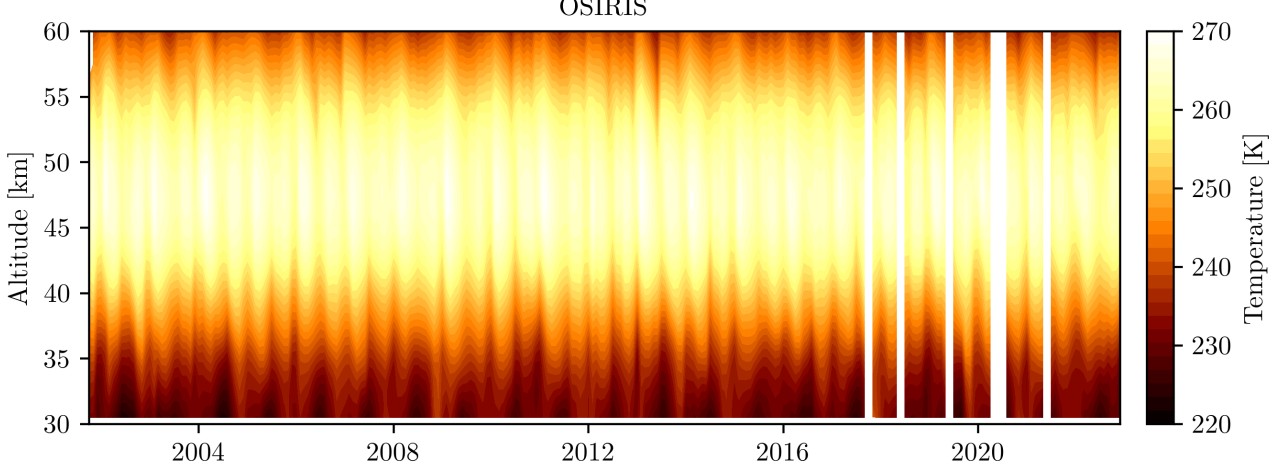

**Figure 1.** Tropical (20° S to 20° N) monthly zonal mean time series calculated from the OSIRIS v7.3 temperature data product.

UV-VIS to retrieve Rayleigh scattering number density, and then use a combination of hydrostatic balance and the ideal gas law to estimate temperatures which goes back to a technique developed for LIDAR sensors by Hauchecorne and Chanin (1980). In limb scatter measurements there is an additional complication of multiple scattering in the atmosphere, the prior methods for OSIRIS and GOMOS both assume that multiple scattering in the atmosphere can be neglected. Chen et al. (2023) retrieved temperatures from 35–75 km with Ozone Mapping and Profiler Suite Limb Profiler (OMPS-LP, Flynn et al., 2004) measure-

ments using a similar technique, however a multiple scattering correction along with altitude normalization was performed to improve the retrieval at lower altitudes.

Recently, the primary OSIRIS data products (ozone, nitrogen dioxide, and stratospheric aerosol) have been updated to v7 and described in several publications (Bognar et al., 2022; Rieger et al., 2019; Dubé et al., 2022). One key improvement in the processing chain is that all of these products are processed self-consistently, rather than separately as was done in the past.

Included in this new v7 processing chain is the retrieval of stratospheric temperature. The profiles are derived through a multi-stage procedure of estimating the Rayleigh scattering signal from OSIRIS spectra, and then integrating hydrostatic balance combined with the ideal gas law. The useful range of these profiles is approximately 35–60 km. The temperature data product is publicly distributed as part of the recently released OSIRIS v7.3 data products. An example of the produced data is shown in Fig. 1. The technique differs from previous techniques in that multiple scattering is included rigorously in the forward model,

with a novel method to estimate the amount of upwelling radiation.

Section 2 describes the OSIRIS measurement technique, with the retrieval technique given in Sect. 3 with associated sensitivity studies and internal assessments in Sect. 4, and lastly intercomparisons with other instruments is given in Sect. 5. A future publication is being prepared which relates the full OSIRIS temperature time series to reanalysis datasets.





## 2 OSIRIS limb scatter measurements

The Swedish satellite Odin (Murtagh et al., 2002) was launched in 2001, and continues to make atmospheric measurements resulting in a more than 22 year ongoing time-series. One of two instruments on-board, OSIRIS, measures limb scattered sunlight in the 280–800 nm spectral region, using a diffraction grating to disperse the signal with a spectral sampling and resolution of about 1 nm. Odin continuously scans the atmosphere in the vertical direction to obtain altitude resolved spectra. Each scan takes approximately 90 s, with a tangent altitude spacing of 2–3 km. Odin primarily operates in three scanning

modes: Strat, Meso, and Strat+Meso. The majority of Odin scans (especially in the recent years) are in the Strat mode, which scans the stratosphere from the surface to $\sim 65$ km.

Odin is in sun-synchronous orbit with a local time at ascending node of $\sim$18:00, which causes the majority of measurements to be made at times near dawn or dusk. Since Odin's orbit is not controlled, the local time has drifted later over time, causing the ascending node to have inconsistent sampling. Therefore in this study we focus only on the descending, or AM, measurements.

On average, this results in $\sim$100 measurements per day across the entire mission.

## 3 Method

Stratospheric temperature is retrieved from OSIRIS limb scatter measurements in a two step procedure. First, Rayleigh scattering number density is inferred from radiances at 310 nm and 350 nm. Next, the number density is used in conjunction with hydrostatic balance and the ideal gas law to obtain temperature. Here we describe the method used to obtain stratospheric

temperature from OSIRIS limb scattered radiances. The temperature retrieval steps are described in reverse order, beginning the process of determining temperature from Rayleigh scattering number density as we believe this is most informative.

### 3.1 Temperature determination

Knowing the Rayleigh scattering number density as a function of altitude, $n(z)$, it is possible to determine atmospheric temperature. The conversion follows almost identically to that of Sheese et al. (2012); Hauchecorne and Chanin (1980); Hauchecorne

et al. (2019), where hydrostatic balance and the ideal gas law are combined to obtain

$$T(z) = \frac{1}{n(z)} \left( n(z_0)T(z_0) - \frac{1}{k} \int_{z_0}^{z} g(z')n(z')m(z')dz' \right), \tag{1}$$

where $T(z)$ is the temperature profile at altitude $z$, $g$ is local gravitational acceleration, $k$ is the Boltzmann constant, $m$ is the mean molecular mass of air, and $z_0$ is a reference altitude. Application of Eq. 1 is straight forward with numerical integration techniques as long as temperature at a reference altitude is known. Sheese et al. (2012) opted to pin the reference altitude at

$\sim$85 km, and calculate the reference temperature from an estimate using OSIRIS A-band emission spectra (Sheese et al., 2010), however, this is only possible for Odin-OSIRIS scans in stratospheric-mesospheric mode. Instead, we calculate the temperature profile using two different reference values at the upper limit of the retrieved Rayleigh scatter number density.





The first reference value is an interpolated MERRA2 (Gelaro et al., 2017) profile used in the standard OSIRIS processing. This value is a reasonable estimate of the mesospheric temperature (Long et al., 2017), and likely produces the most accurate OSIRIS temperature profile, however, any spurious trend in the MERRA2 data at the reference altitude will be aliased into the OSIRIS temperature values. To obtain more robust trend estimates, we compute a second temperature profile pinned by a climatological value estimated from the NRLMSISE-00 model (Picone et al., 2002). Differences in these two temperature products are assessed in the following sections.

There are two natural ideas to take note of with Eq. 1. First, the temperature profile does not depend on the magnitude of the retrieved Rayleigh scattering number density, only its shape. In other words, scaling the retrieved number density profile by a constant factor does not change the retrieved temperature profile. The scaling invariance provides some robustness to errors either in the instrument calibration or radiative transfer modelling. The second is that errors in the reference temperature directly introduce errors in the retrieved profile, however, this errors drops off exponentially in altitude and can be immediately estimated.

## 3.2 Rayleigh scattering number density retrieval

Rayleigh scattering number density is retrieved using a standard optimal estimation retrieval scheme (Rodgers, 2000). The state vector, $\boldsymbol{x}$, consists of the logarithm of dry air number density on an altitude grid and is updated iteratively through the relation

$$\boldsymbol{x}_{i+1} = \boldsymbol{x}_i + \left[\mathbf{K}^T\mathbf{S}_y^{-1}\mathbf{K} + \boldsymbol{\Gamma}^T\boldsymbol{\Gamma}\right]\mathbf{K}^T\mathbf{S}_y(\boldsymbol{y} - F(\boldsymbol{x}_i)), \tag{2}$$

where $\boldsymbol{y}$ is OSIRIS measured radiance at 350 nm, $\mathbf{S}_y$ is the error covariance matrix of these measurements, $\boldsymbol{\Gamma}$ is a second order Tikhonov regularization matrix, $\mathbf{K}$ is the Jacobian matrix, and $F$ is a combination of an OSIRIS instrument model and the SASKTRAN radiative transfer model (Zawada et al., 2015). Included in the forward model calculation are the OSIRIS v7 estimates of stratospheric aerosol, ozone, and nitrogen dioxide. The Jacobian matrix is calculated analytically by the SASKTRAN radiative transfer model (Zawada et al., 2017). The logarithm of dry air number density is used since its second derivative is approximately zero inside the Earth's atmosphere.

The state vector is represented on a 1 km uniform altitude grid beginning at 30 km, and extending to the top of the OSIRIS scan, or 80 km, whichever is lower. We expect that any measurement below ∼35 km is contaminated by the effects of stratospheric sulfate aerosol, setting the lower-bound at 30 km allows us to test this assumption without adversely affecting the entire profile. Most of the time OSIRIS is operating in stratospheric mode where each scan stops at ∼65 km, in these cases the scan top is set as the upper limit of the state vector. OSIRIS can also operate in a mesospheric mode, where the scan extends to ∼100 km or above, here the upper limit is set to 80 km to avoid regions of significant stray-light. Above the upper-bound and below the lower-bound the profile is scaled by a constant factor to line up with the state vector each iteration.

The second order Tikhonov factor is chosen in a semi ad-hoc fashion so that the average $\chi^2$ value, calculated through,

$$\chi^2 = \frac{(\boldsymbol{y} - F(\hat{\boldsymbol{x}}))^T\mathbf{S}_y(\boldsymbol{y} - F(\hat{\boldsymbol{x}}))}{M - 1}, \tag{3}$$




where $M$ is the number of measurements, is approximately equal to 1. This criteria minimizes the amount of overfitting done by the retrieval, and results in a mean vertical resolution of 3–3.5 km (see Sect. 4.3.4).

### 3.3  Lambertian equivalent reflectance estimation

Radiances at 350 nm contain significant signal from multiple scattering. Multiple scattering in this wavelength region is primarily an upwelling radiation effect, light scatters either at the ground or the cloud top, and undergoes multiple scattering events until scattering one final time along the instrumental line of sight. Accounting for every physical process in this scattering path is not feasible both from a lack of atmospheric composition information, and also the large increase in radiative transfer modelling complexity. Instead, we assume a clear sky troposphere, and a Lambertian reflectance at the Earth's surface parameterized by a scalar, spectrally invariant, surface albedo. Prior to beginning the Rayleigh scattering number density retrieval we estimate the surface albedo.

For the OSIRIS v7 retrievals of ozone, stratospheric aerosol, and nitrogen dioxide, surface albedo is required and retrieved, however this estimate not suitable for use in the temperature retrieval. The primary surface albedo estimated is retrieved at 675 nm to be applicable for wavelengths in the aerosol retrieval and Chappuis ozone absorption bands, however the albedo at this wavelength may be significantly different than that at 350 nm. The method used also can not simply be repeated at 350 nm. Absolute radiances at a high altitude can be used to estimate the surface albedo, however these radiances at 350 nm are already used in the retrieval of Rayleigh scattering number density, and do not contain any independent pieces of information for surface albedo. Instead, a new method has been developed to estimate surface albedo in this region.

Our goal is to obtain a piece of information in the OSIRIS radiances that varies with surface albedo but is approximately independent of Rayleigh scattering number density and ozone absorption, which are the primary radiative effects in this wavelength region. Simulations were performed in the nearby spectral area using varying amounts of ozone and surface albedo at an altitude of 60 km and are shown in Fig. 2. At 350 nm, the spectrum is only sensitive to the underlying surface albedo. From 310 nm to 340 nm, we see dependence on the atmospheric ozone profile. However this dependence is not through absorption along the line of sight, rather it is attenuation of the multiply scattered upwelling signal. Direct line of sight sensitivity to ozone begins around $\sim 295$ nm and continues to increase as wavelength decreases. The spectral range of approximately $\sim 295$ nm to $\sim 305$ nm is generally insensitive to both the atmospheric ozone profile and the surface albedo.

To retrieve effective surface albedo we define the measurement vector,

$$y_a = \log I(350\,\text{nm}, 60\,\text{km}) - \log I(305\,\text{nm}, 60\,\text{km}), \tag{4}$$

which is approximately insensitive to both Rayleigh scattering number density and ozone absorption, but retains sensitive to the surface albedo. The retrieval then proceeds by setting the state vector to be the scalar surface albedo, and performing an unregularized form ($\Gamma = 0$) of Eq. 2. Since SASKTRAN is unable to analytically compute the derivative of surface albedo, we calculate the measurement vector once with an albedo of 0 and estimate the derivative as,

$$\frac{\partial I}{\partial a} = K = \frac{y_a(a) - y_a(0)}{a}. \tag{5}$$





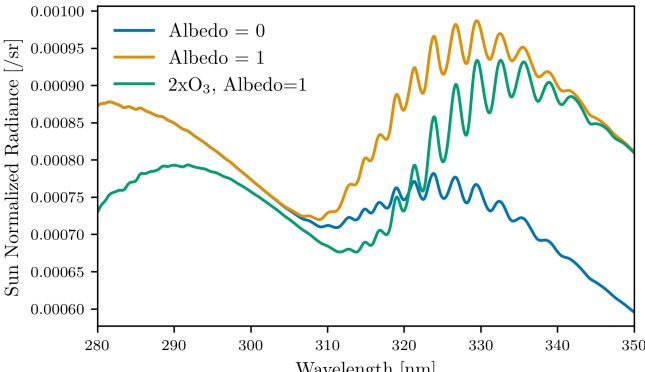

**Figure 2.** Sun normalized radiance spectra as simulated by SASKTRAN at $60\,\mathrm{km}$ for three different scenarios. "Albedo 0": Standard atmosphere, surface albedo set to 0. "2 x $O_3$": Atmosphere with twice as much ozone, albedo set to 1. "Albedo: 1": Standard atmosphere, surface albedo set to 1.

Using an approximate Jacobian likely increases the number of iterations required in the retrieval, but reduces computation time overall since the problem is sufficiently linear.

We have noticed that many scenes result in the retrieval wanting to push the albedo negative. Rather than allowing a negative albedo in SASKTRAN, we leave the albedo at the minimum value and add a flat absorber to the troposphere. The absorber is assumed to have a constant number density from $0$–$5\,\mathrm{km}$, and have a spectrally flat cross section. The state vector element is then switched from albedo to vertical optical depth of this absorber. The Jacobian is estimated in a fashion similar to Eq. 5 using the previously computed radiance at an albedo of zero. Currently any scene where an absorber is added is recommended to be filtered out.

## 4 Internal assessment and sensitivity studies

### 4.1 Absolute calibration effect

As previously noted, a scaling of the retrieved Rayleigh scatter number density does not directly influence the retrieved temperature profile. However, a multiplicative scaling of the radiance profile does not directly result in a scaling of the number density profile due to both the non-linear nature of the radiative transfer problem and the change in multiple scatter intensity as a function of altitude.

Little effort has been put into characterizing the absolute calibration of OSIRIS during the multi-decade mission. Most other retrieved OSIRIS data products (ozone, stratospheric aerosol, nitrogen dioxide) are only weakly sensitive to the absolute calibration through coupling with the estimated surface albedo. The platform itself also contains no on-board calibration sources and is unable to directly measure known brightness features that fully illuminate the slit. An updated OSIRIS absolute calibration has been developed specifically to improve the temperature retrieval data product.



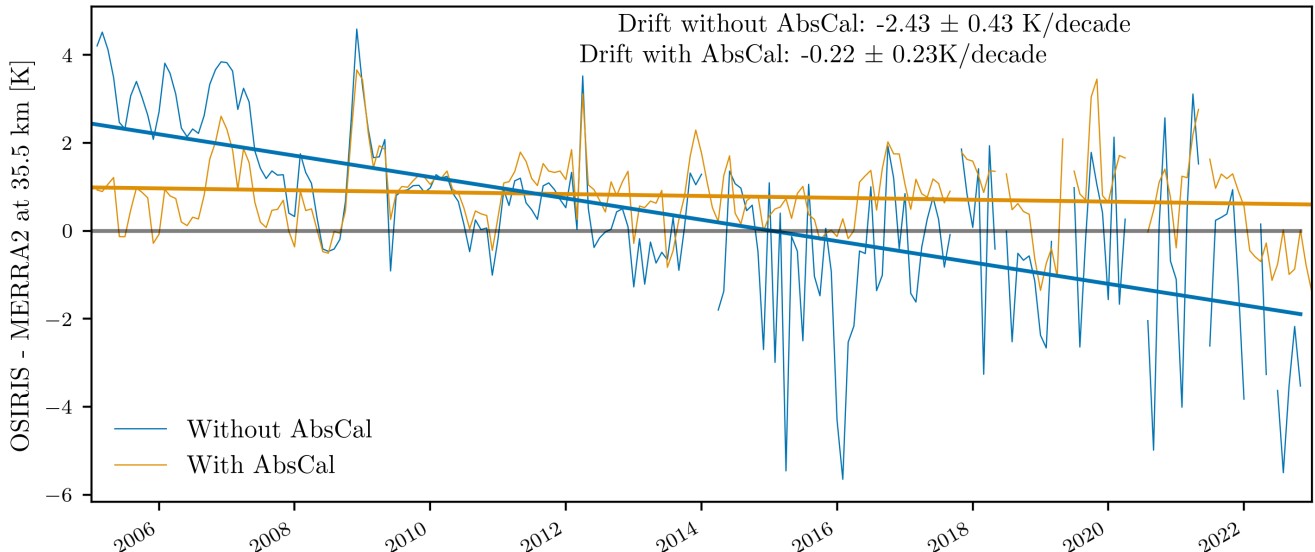

**Figure 3.** Monthly zonal mean differences between OSIRIS and MERRA2 at 35.5 km in the 10° S to 10° N latitude band with and without the new absolute calibration. Linear fits are shown to each time series with associate $2\sigma$ error estimates.

OSIRIS measured radiances are compared to SASKTRAN simulations using reanalysis atmospheric inputs over time for specific measurement geometries that we are confident can be simulated accurately. We restrict ourselves to scans with a tangent point solar zenith angle of ~90 degrees which minimizes the amount of upwelling radiation, and solar scattering angles of ~90 degrees which limits the change in solar zenith angle along the line of sight. Larger solar zenith angles may

further increase the relative importance of single scattering, however solar zenith angles above 90 degrees result in the solar beam passing through altitudes below the measurement tangent point which is undesired.

As an initial assessment of the new absolute calibration we have compared the v7.3 temperature data product to an older research product that did not contain the absolute calibration correction in Fig. 3. We analyze the lowest usable altitude (35.5 km) since low altitudes are both more-so affected by errors in absolute calibration (see Sect. 4) and are expected to be more accurate

in MERRA2. Without the absolute calibration, a significant (-2.43 ± 0.43 K/decade) drift is observed relative to MERRA2. With the new absolute calibration the drift becomes insignificant (-0.22 ± 0.23 K/decade). Potential drifts at low altitudes are further examined in Sect. 5.4.

## 4.2   Aerosol contamination at low altitudes

The limiting factor in pushing the retrieval lower into the stratosphere is the presence of stratospheric aerosol scattering. It
is difficult to decouple the scattering effects of aerosols from that of Rayleigh scattering particles. The temperature retrieval includes the OSIRIS v7 retrieved aerosol extinction as an ancillary parameter, however this retrieval primarily uses information from longer wavelengths (Rieger et al., 2019). Uncertainties in the aerosol particle size distribution and composition can



introduce significant errors in the shorter wavelengths that are used in the temperature retrieval. Rather than setting the retrieval lowerbound at a safe altitude where we are certain there is no aerosol influence (e.g. 40 km), we have opted to set the
lowerbound to an altitude that we know will be contaminated with aerosol (30 km), and then post-filter the data.

Since the OSIRIS v7 aerosol profile is used as ancillary data inside the retrieval, it is not the presence of aerosol itself that causes errors in the retrieval, but rather uncertainties due to unknown aerosol particle size and composition. Therefore, we determine which altitudes are adversely affected by aerosol uncertainties by analyzing correlations to the single scattering angle (SSA). Similar ideas have been used to analyze errors of unknown aerosol particle size on retrieved aerosol extinction.
Figure 4 shows mean tropical monthly zonal mean differences relative to the MERRA2 ancillary data as a function of single scattering angle for different altitude levels. At the lowest retrieved altitude (30.5 km), we see a strong dependence on single scattering angle, with temperature differences varying almost linearly from 5 K to -5 K over the 50° of observed OSIRIS scattering angles. The dependence decreases with increasing altitude, as expected with decreasing aerosol concentrations. At 35.5 km the observed scattering angle dependence reduces to ±0.5 K, and stays under that level for all altitude levels above it.
Therefore we recommend that the retrieved temperature data product not be used at altitude levels below 35 km.

### 4.3 Error analysis

We have identified three primary sources of error that influence the retrieved temperature data product. The first is errors due to errors in the assumed reference temperature. The second is errors due to random noise present within the OSIRIS spectral measurements. The third is error from errors in the absolute signal level of the measurements, primarily originating from errors
in the instrumental absolute calibration. To assess these errors we processed 100 OSIRIS orbits from the year 2009 chosen randomly with different retrieval configurations. The following subsections outline the setup of these tests and highlight the results.

### 4.3.1 Reference temperature errors

Errors due to errors in the reference temperature are estimated by comparing the retrieved temperature that uses NRLMSISE-00
as the reference temperature to the retrieved temperatures pinned to the MERRA2 value. Differences between these two values provides a conservative estimate for uncertainty at the reference temperature. The mean differences observed for the 100 test orbits is shown in Fig. 5. Near the reference altitude, both the mean difference and observed scatter approach 5 K, decreasing exponentially in altitude.

The exponential decrease in altitude can be directly seen from Eq. 1. An error in the pinning temperature, $\delta T(z_0)$, directly
results in an error on the retrieved temperature profile,

$$\delta T(z) = \frac{n(z_0)\delta T(z_0)}{n(z)}. \tag{6}$$

At the nominal reference altitude, 65 km, we observe mean differences of 5.6 K between the climatological NRLMSISE-00 value and the interpolated MERRA2 temperatures, with a standard deviation of 5.1 K. The error on the retrieved profile can be estimated directly from the retrieved number density, but owing to the exponential nature of the atmospheric density profile it





**Figure 4.** Differences between OSIRIS and MERRA2 at a set of tangent altitudes as a function of single scattering angle in the $10°$ S to $10°$ N latitude band. Slopes are shown with associated $2\sigma$ error estimates.





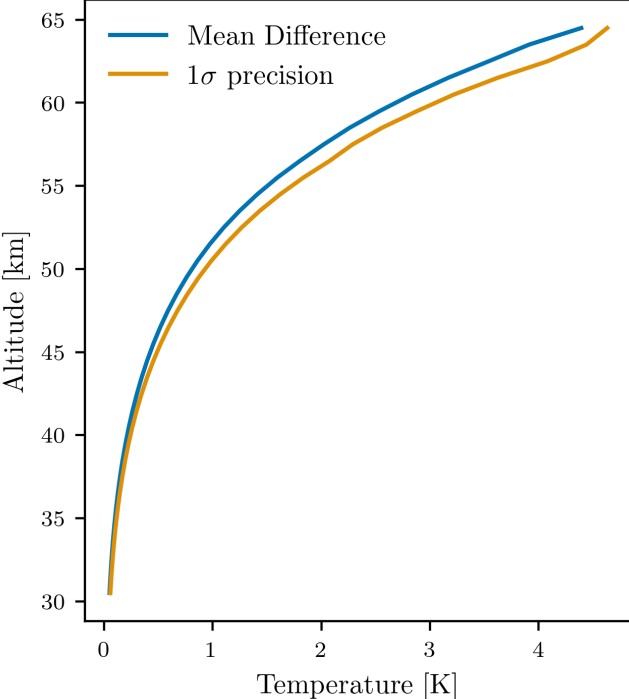

**Figure 5.** Mean differences and $1\sigma$ standard deviation between the NRLMSISE-00 pinned and MERRA2 pinned temperature data products for the 100 test orbits.

is well approximated through the relation,

$$\delta T(z) = \exp\left(-\frac{z_0 - z}{z_s}\right) \delta T(z_0), \tag{7}$$

where $z_s$ is the scale height of the atmosphere ($\sim 8.5\,\mathrm{km}$). From this relation we can see that the error due to errors in the reference temperature decreases exponentially in altitude, and for a reference altitude of $65\,\mathrm{km}$, the error reaches a 10-fold reduction at approximately $45\,\mathrm{km}$.

### 4.3.2 Random noise

We assess the random noise component in two, somewhat equivalent, ways. The random noise component of the retrieval can be estimated through standard linear error analysis, for details on how this is applied to the derived temperature product see Sect. A. The OSIRIS signal to noise ratio (SNR) is approximately $\sim 500$ for the wavelength used in the retrieval, and has minimal variation in altitude or viewing condition since an auto-exposure time process is used during nominal operation. Since 215 almost the entire signal originates from Rayleigh scattering, the precision on the retrieved number density, and the temperature values, are expected to have precision very close to this SNR value.



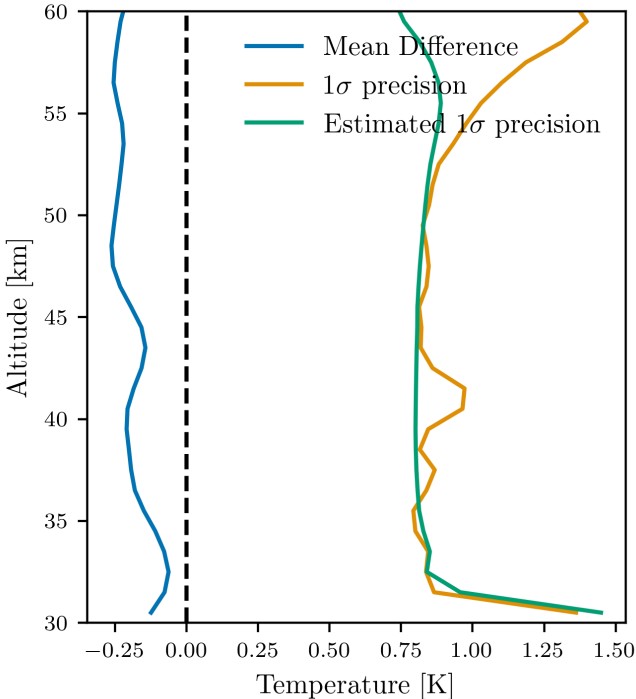

**Figure 6.** Mean differences and $1\sigma$ standard deviation for the 100 test orbits when the nominal retrieval wavelength is moved to 345 nm from 350 nm.

The second method to determine the random component of the retrieval is to change the wavelength used to retrieved Rayleigh scattering number density. For this test, we move the retrieval wavelength to 345 nm from 350 nm, with the results shown in 6. The observed scatter matches closely with the predicted scatter, and is a constant 0.8 K from 35 km to 55 km, increasing slightly to values slightly greater than 1 K at higher altitudes. We believe the slight deviation between the two methods at 60 km is caused by pseudo-systematic errors in the radiance at the upper retrieval limit, most likely stray light. A small -0.2 K mean bias is observed when moving the wavelength to 345 nm. The cause of the bias is unknown, but does suggest that there is the potential for small, likely constant in time, biases in the retrieved temperature depending on the exact OSIRIS wavelength chosen.

### 4.3.3 Absolute calibration error

The temperature retrieval uses two distinct wavelengths, 350 nm and 305 nm. Errors in the absolute calibration of each of these channels will alias into errors in the retrieved temperature values. However, based on the technique to determine the absolute calibration described in Sec 4, we expect errors in the estimated absolute calibration to be highly correlated between these





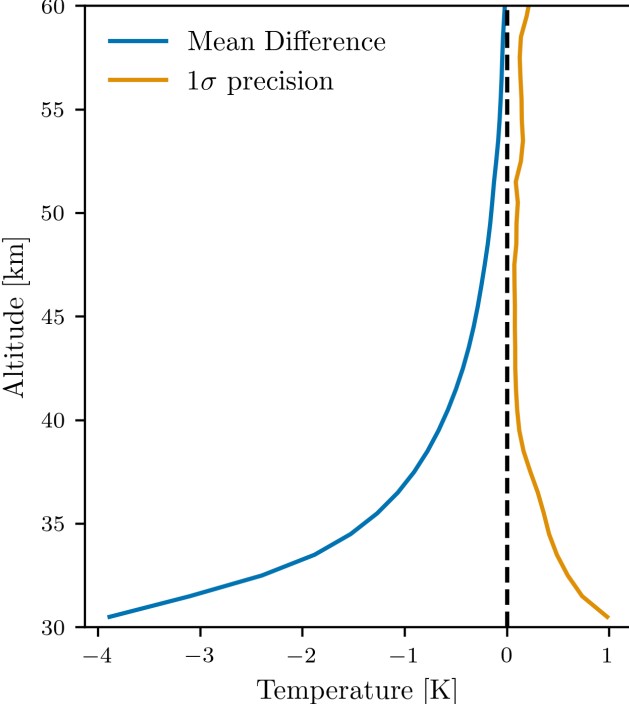

**Figure 7.** Mean differences and $1\sigma$ standard deviation for the 100 test orbits when the OSIRIS absolute radiances are artificially scaled by 1.05.

two measurements. Therefore to assess errors in the absolute calibration, we artificially multiply the observed radiance by a
spectrally flat value of 1.05. The results of the test are shown in Fig. 7.

There is a small increased scatter from the absolute calibration scaling, however above 35 km it is less than 0.3 K, and likely
is a result of the non-linear nature of the problem. The bias introduced from the absolute calibration scaling appears to increase
exponentially with decreasing altitude, reaching -1 K at 37.5 km. At the lowest usable altitude of the retrieval, 35 km, the mean
retrieved temperature difference nears -2 K for a 5% change in absolute radiance.

**4.3.4   Other potential forms of bias**

While the previous three sections outline what we believe to be the primary sources of error in the retrieved temperature data
product, there do exist other sources of error for the retrieved data product. In addition, we have tested the retrieval sensitivity
to:

– The assumed aerosol particle size in the aerosol retrieval by increasing the log-normal median radius from 80 nm to
240       100 nm

– The neglect of polarization in the retrieval forward model by including the effects of polarization in the calculation.





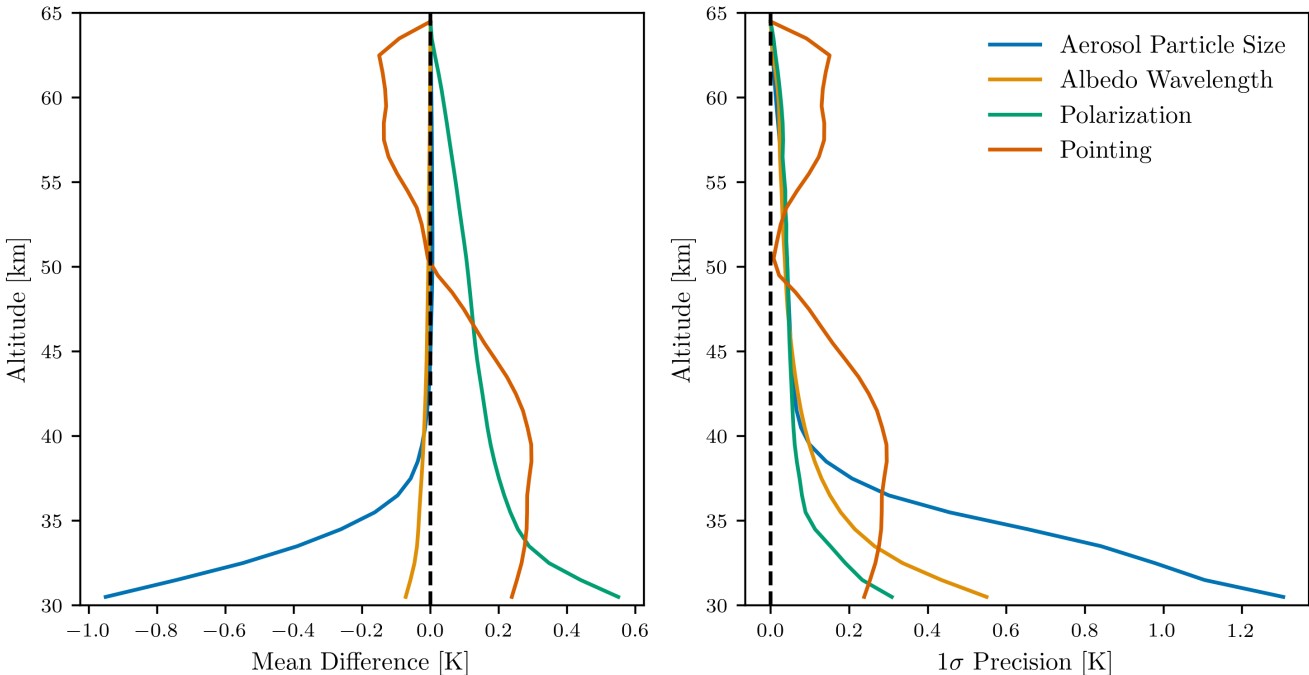

**Figure 8.** Mean differences and $1\sigma$ standard deviation for the 100 test orbits for perturbations in aerosol particle size, albedo retrieval wavelength, and simulating polarization in the retrieval forward model. See text for details.

- The wavelength used in the equivalent Lambertian reflectance retrieval by switching the wavelength from 305 nm to 307 nm

- A pointing knowledge shift of 100 m at the tangent point.

The results of these studies are shown in Fig. 8.

Generally, all four effects introduce very little bias on the retrieved temperature data product. Aerosol particle size and the wavelength used for the albedo retrieval are negligible above 35 km. The neglect of polarization and potential pointing errors have the largest bias, introducing a bias on the order of 0.2 K above 35 km. We note that polarization errors are partly cancelled from the absolute calibration correction. A 100 m pointing shift causes biases of ±0.25 K depending on the direction of the shift

and whether or not the altitude is above/below the stratopause. The scatter of the aerosol particle size, albedo wavelength, and polarization effects are less than 0.1 K above 40 km, and rapidly increases below that. The aerosol particle size effect is largest, increasing to 0.6 K at 35 km. Owing to the large uncertainties in stratospheric aerosol composition, it is not unreasonable to expect that errors larger than this could be observed in the 35–40 km altitude region in time periods of enhanced aerosol.



### 4.3.5 Vertical resolution

The vertical resolution of the Rayleigh scattering number density retrieval is available through the retrieval averaging kernel, however the averaging kernel for the temperature data product is not the same due to the conversion in Eq. 1. Note that since the temperature conversion is non-linear, there is no exact conversion from the Rayleigh scattering number density averaging kernel to the temperature averaging kernel. The proper way to apply the averaging kernel to a comparison dataset would be to convert to number density, apply the averaging kernel, then re-apply Eq. 1. However since the temperature at a specific level is primarily influenced by number density at that level (see Eq. A9), the vertical resolution of the number density retrieval provides a rough estimate of the vertical resolution of the temperature data product.

For each retrieved profile the vertical resolution is estimated through fitting a Gaussian shape to the central peak of the averaging kernel. Generally mean vertical resolutions are in the range 3.0–3.5 km across the usable retrieval altitudes, which is on the same order as the OSIRIS vertical sampling (2–3 km). A summary of the vertical resolution as a function of altitude is provided in Table 1.

### 4.4 Summary and discussion

All of the assessed sources of error and their quadrature summation are shown in Fig. 9. Bias is dominated by biases in the reference temperature above 45 km, and then becomes dominated by errors in the absolute calibration below that. Biases are on the order of 1–4 K depending on the altitude range considered. It should be remembered that the estimated error due to absolute calibration was calculated using an assumed error of 5 % in the absolute radiances, which is only a reasonable estimate. The precision of the retrieval is controlled by random noise below 50 km, and then dominated by noise in the reference temperature above that. Precision is typically on the order of 1 K up to 50 km, and then increases to 5 K at the highest altitudes. A summary of the estimated precision, bias, vertical resolution, and useful notes to consider for prospective data users is provided in Table 1.

So far, the effect of polar mesospheric clouds (PMCs) has not been discussed. PMCs can have a significant impact on the limb scattered radiance signal, and can cause extreme variations in the retrieved temperature data product (Chen et al., 2023). Since typical OSIRIS scans terminate at 65 km, well below the nominal altitude of PMCs (80-85 km), accurate detection and filtering of PMCs in OSIRIS data is challenging. Therefore we recommend that use of data in the high latitude summer months be avoided.

## 5 Comparison with other datasets

### 5.1 Data descriptions

The retrieved OSIRIS temperatures were validated through comparison with temperature profiles from the Microwave Limb Sounder (MLS, Waters et al., 2006) and the Atmospheric Chemistry Experiment-Fourier Transform Spectrometer (ACE-FTS,





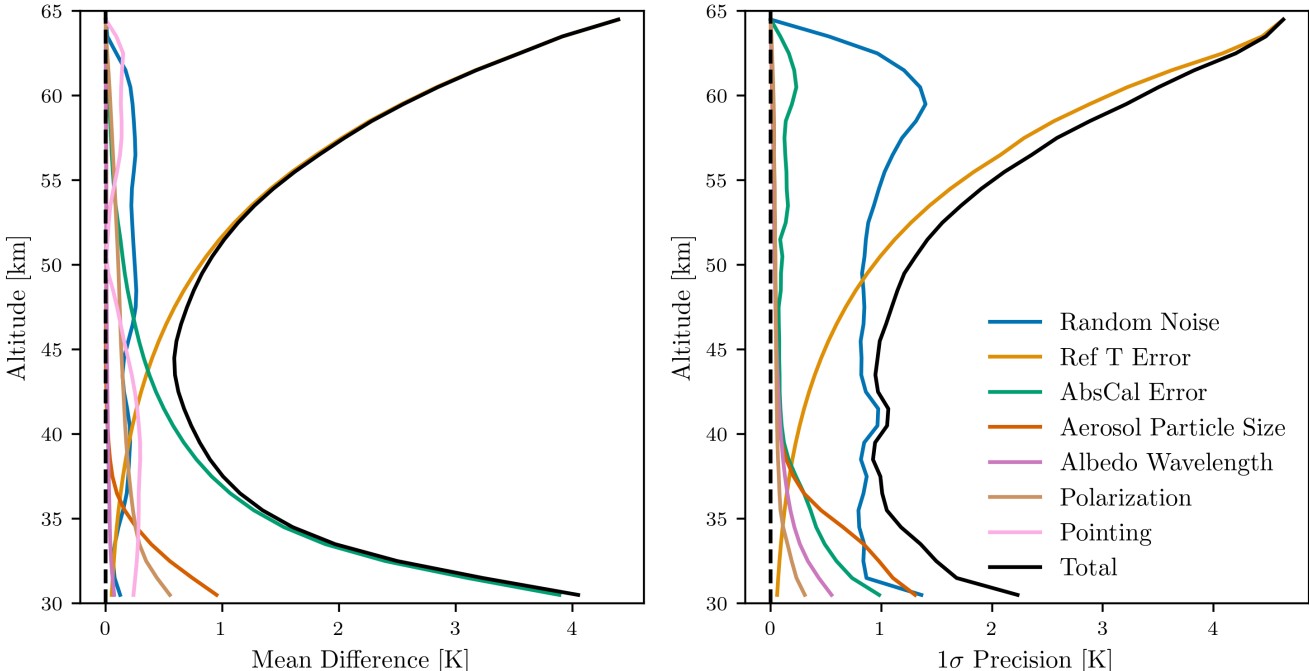

**Figure 9.** Mean differences and $1\sigma$ standard deviation for the 100 test orbits for all perturbation studies performed. See text for details.

**Table 1.** Summary of the estimated precision, bias, and vertical resolution of the temperature data product for each altitude range.

| Altitude Range | Estimated Precision ($1\sigma$) | Estimated Bias | Vertical Resolution (FWHM) | Notes |
|---|---|---|---|---|
| 30-35 km | 1–2 K | 1–4 K | 3.0 km | Not suitable for scientific use due to aerosol contamination |
| 35-40 km | 1 K | 1–2 K | 3.0 km | Errors dominated by absolute calibration and systematics. Caution should be taken in time periods following extreme stratospheric aerosol activity. |
| 40-45 km | 1 K | 1 K | 3.0 km | |
| 45-50 km | 1–1.5 K | 1 K | 3.0 km | |
| 50-55 km | 1.5–2 K | 1–2 K | 3.3 km | |
| 55-60 km | 2–3 K | 2–3 K | 3.5 km | Errors dominated by reference temperature errors |
| 60-65 km | 3–4 K | 3–5 K | 3.5 km | Heavily influenced by reference temperature |





Bernath et al., 2005). All comparisons use the version of the OSIRIS temperatures estimated with MERRA2 as the upper
altitude reference value.

MLS observes microwave limb emissions, and measures approximately 3500 vertical profiles each day. Temperatures are retrieved near the $O_2$ spectral lines at 118 GHz and 239 GHz (Livesey et al., 2022). Version 5 of the retrieval is used here, and all profiles are filtered according to the guidelines provided in Livesey et al. (2022). The MLS geopential height (GPH) profiles that are retrieved along with each temperature profile are used to convert the temperature profiles from a vertical pressure grid
to a geometric altitude grid before comparing to OSIRIS.

ACE-FTS measures approximately 30 atmospheric transmission profiles each day using a solar occultation geometry: there are ~15 profiles at sunrise and ~15 profiles at sunset. ACE-FTS temperatures are retrieved using observations of $CO_2$ spectral features in the mid-infrared. Profiles from version 4.2 of the temperature retrieval, described in Boone et al. (2020), are considered here. The observations are filtered according to the data quality flags developed by Sheese et al. (2015) before being
used.

## 5.2   Coincident comparisons

Coincident profiles between OSIRIS and each of MLS and ACE-FTS are compared. The coincidence criteria were chosen such that there were a reasonable number of profiles in each of the Northern Hemisphere mid-latitudes, tropics, and Southern Hemisphere mid-latitudes, but without the pairings being too far apart in space and time. For OSIRIS and MLS the coincident
profiles are within 6 hours, 2 degrees latitude, and 5 degrees longitude. This largely corresponds to comparing OSIRIS, with a local time near 6:30 AM, to nighttime MLS profiles. For OSIRIS and ACE-FTS the coincident profiles are within 3 hours, 5 degrees latitude, and 10 degrees longitude. The 3 hour time criteria means that only ACE-FTS sunrise profiles are considered.

The mean biases between OSIRIS and MLS temperature profiles in three broad latitude bins are given by the blue lines in Figure 10, while the mean biases between OSIRIS and ACE-FTS are given by the green lines. In general OSIRIS and ACE-FTS
agree within 3 K at all latitude and altitudes, with OSIRIS biased high compared to ACE-FTS. Differences slightly exceed the predicted systematic biases in Sect. 4.3.5.

MLS and OSIRIS agree within 5 K at all latitudes and altitudes below 55 km. The sign of the bias varies with latitude and altitude. (Schwartz et al., 2008) showed that a previous version of the MLS temperature profiles were biased low by up to 10 K compared to numerous other datasets above 1 hPa. This is likely the main cause of the bias between MLS and OSIRIS above
50 km.

A possible explanation for the observed biases is from the diurnal variation of temperatures, i.e., tidal effects. A correction was performed to assess the impact of the coincident profile sampling on the observed biases. The sampling correction was calculated as the difference between MERRA2 temperatures interpolated to the OSIRIS profile geolocation and MERRA2 temperatures interpolated to the ACE-FTS or MLS geolocation. Since MERRA2 has a three hour temporal resolution, tidal
effects are not well-resolved, however it gives an indication of whether or not the differences are potentially explainable by tides. These results are shown respectively by the orange and red lines in Figure 10. The effect of sampling is at most 1 K for ACE-FTS and 4 K for MLS. Since only ACE-FTS sunrise profiles are used, and OSIRIS usually measures close to sunrise, the





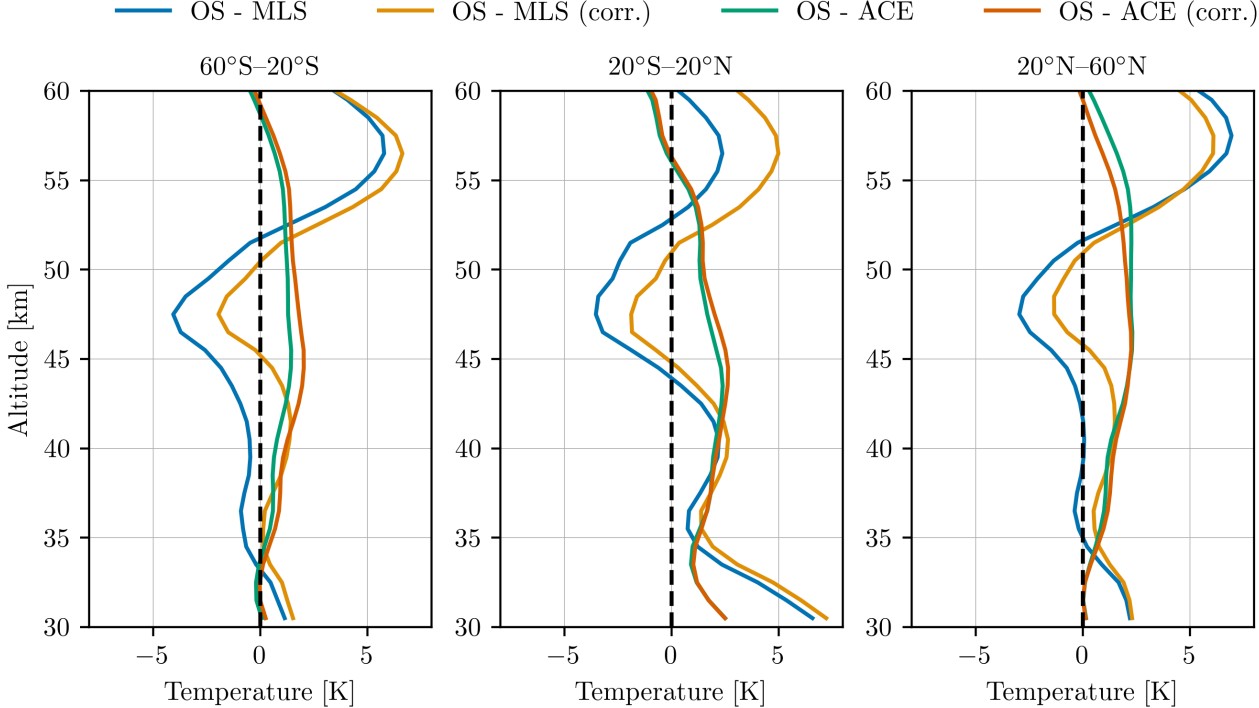

**Figure 10.** Mean difference between coincident temperature profiles retrieved from OSIRIS, MLS, and ACE-FTS in three latitude bands.

effect is generally negligible. For MLS, the sampling effect can be as large as 1–3 K depending on the exact area, suggesting that some of the biases observed between MLS and OSIRIS are likely due to diurnal sampling.

## 5.3 Seasonal cycle evaluation

The primary driver of longer-scale temperature variations in the middle atmosphere is through the seasonal cycle, and thus validation of the seasonal cycle serves as a good test that the retrieval is working as expected. The mean seasonal cycles in temperatures from OSIRIS, MLS, and ACE-FTS are compared in Figure 11. The structure of the seasonal cycle is broadly similar for each dataset. The greatest difference occurs in the Southern hemisphere, where MLS and ACE-FTS display colder winter temperatures than OSIRIS. The ACE-FTS and MLS temperatures are also slightly colder than those from OSIRIS in the Northern hemisphere winter.



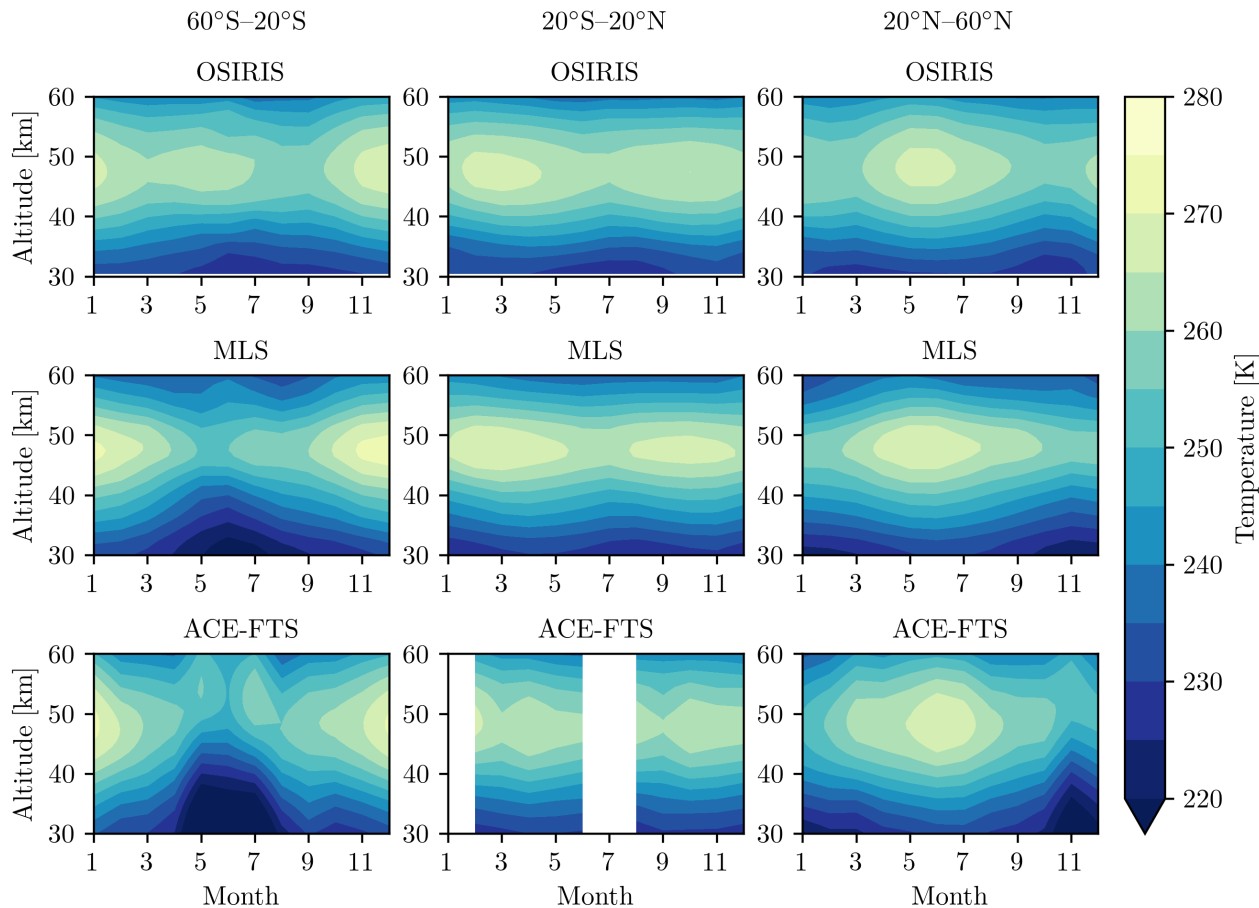

**Figure 11.** Mean seasonal cycle in temperature for each of OSIRIS, MLS, and ACE-FTS in three latitude bands.

## 5.4 Time series comparisons

The deseasonalized time series for each of the three datasets is shown in Fig. 12. Overall there is very good agreement between the variability of the three datasets at each latitude and altitude. There are larger variations in the ACE-FTS temperatures compared to MLS and OSIRIS, particularly in the mid-latitudes, which could be because of ACE's less-dense sampling.

To assess possible drifts in the dataset, in Fig. 13 a linear fit has been performed on the differences in anomalies between OSIRIS and ACE-FTS, and OSIRIS and MLS. For the majority of bins the observed drift between OSIRIS and ACE/MLS is not statistically significant. When there is a statistically significant drift it is usually less than $\pm 1$ K/decade and often does not show up in the same place with respect to ACE/MLS. A consistent cooling greater than 1 K/decade is observed relative to ACE in the middle southern hemisphere, however since it does not show up relative to MLS and ACE sampling is poor in this latitude band we expect it is a sampling artifact. In the 45–50 km region at all latitudes a consistent warming is seen relative to MLS



**Figure 12.** Monthly mean deseasonalized temperature anomalies for OSIRIS, ACE-FTS, and MLS. Results are provided in three latitude bands and 10 km intervals.



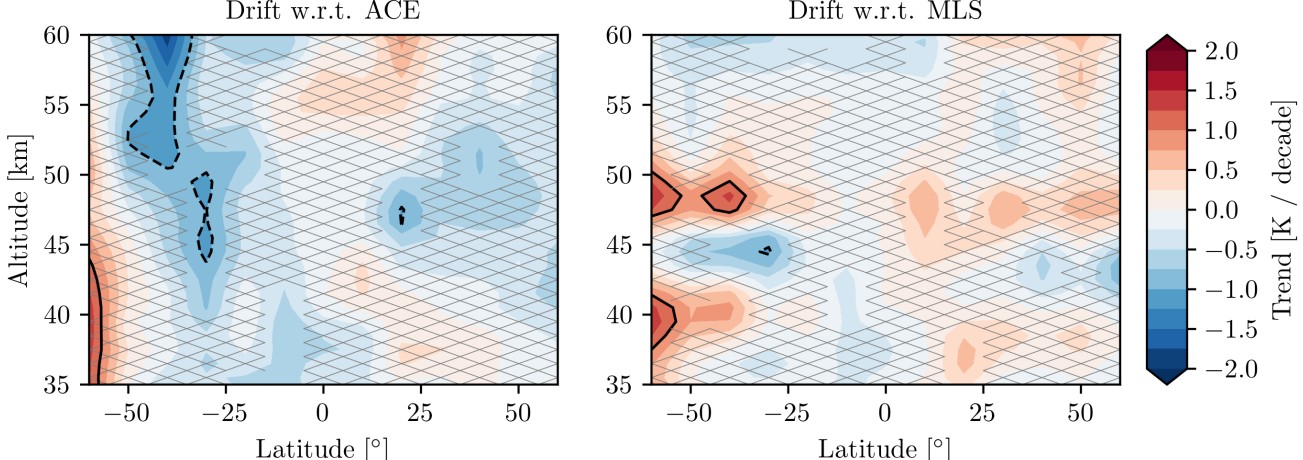

**Figure 13.** Trend in the difference between monthly mean deseasonalized temperature anomalies for OSIRIS−ACE (left) and OSIRIS−MLS (right). Trends are calculated using a simple linear fit. The 1 K and -1K contours are marked in black. Hatched regions denote a statistically insignificant trend.

that does not appear when compared to ACE-FTS. Given the proximity of this altitude region to the stratopause, it is possible that this is an artifact of the coarse vertical resolution of MLS in this region (∼5 km) and the conversion of pressure levels to altitude levels using geopotential height. Overall, with the exception of the 35–42 km region at 60°S, there are no areas of

consistent drifts over 1 K/decade between OSIRIS and MLS/ACE. The absence of latitudinally constant drifts at low-altitudes suggests that the absolute calibration correction from Sect. 4.3.3 is working as intended.

# 6 Conclusions

The method to retrieve atmospheric temperatures with a useful range of 35–60 km included as part of the OSIRIS v7.3 processing has been described. The method combines a retrieval of Rayleigh scattering number density with hydrostatic balance

and the ideal gas law to recursively estimate temperature beginning with a reference temperature at a reference high altitude. In contrast to other techniques, multiple scattering is handled directly in the retrieval, with an equivalent Lambertian surface reflectance estimated using an absorbing wavelength. A detailed error analysis has been performed, estimated precisions to be 1–4 K, with potential biases of 1–5 K, and a vertical resolution in the range of 3–3.5 km. Errors in the range 45–60 km are dominated by uncertainties in the reference temperature used to pin the solution, and errors at altitudes below 45 km are

controlled by uncertainties in the absolute calibration of the instrument. An absolute calibration correction has been developed for OSIRIS which greatly reduces drifts in the time series.

Comparisons to temperatures from ACE-FTS and MLS show very good agreement, with biases less than 5 K at most latitudes and altitudes. Seasonal cycles are generally consistent between OSIRIS, MLS, and ACE-FTS, but are challenging to effectively compare because of differences in sampling and local time coverage. Anomaly comparisons between the three instruments



show similar variability. Drifts have been assessed between OSIRIS and ACE-FTS/MLS, and in most areas are statistically insignificant, and rarely exceed 1 K/decade.

*Data availability.*

The produced OSIRIS temperature data record can be obtained at Zawada et al. (2023). Other data sources used in this publication are:

– ACE-FTS data quality flags (Sheese and Walker, 2022)

   – ACE-FTS data are available at https://databace.scisat.ca/level2/ (ACE-FTS, 2022)

   – MLS temperature data from https://disc.gsfc.nasa.gov/datasets/ML2T_005/summary (Schwartz et al., 2020b)

   – MLS GPH data from https://disc.gsfc.nasa.gov/datasets/ML2GPH_005/summary (Schwartz et al., 2020a)

## Appendix A: Numerical conversion from Rayleigh scattering number density to temperature

The conversion to temperature begins from the equation of hydrostatic balance,

$$p(z) = p(z_0) - \int_{z_0}^{z} g(z')n(z')m(z')dz', \tag{A1}$$

or equivalently,

$$p_{i+1} - p_i = \Delta p_i = - \int_{z_i}^{z_{i+1}} g(z')n(z')m(z')dz', \tag{A2}$$

where the index $i = 0$ represents the top (pinning) altitude, with layers descending vertically, and $p$ is pressure. From this form

we can begin with the reference $p_0$ value, and recursively compute the temperature downwards through the atmosphere.

To evaluate the integral, we use the fact that SASKTRAN internally performs linear interpolation of the number density between grid points, then, within a single layer the number density is,

$$n_i(z) = n_i + \frac{n_{i+1} - n_i}{z_{i+1} - z_i}(z - z_i). \tag{A3}$$

We assume the molar mass of air is constant as a function of height, and given as 28.97 g/mol, and that gravitational acceleration

decreases quadratically with height according to the relation,

$$g(z) = g_0 \frac{R_e^2}{(R_e + z)^2}, \tag{A4}$$



where $R_e$ is the radius of the Earth computed from the average ground track location for each OSIRIS scan, and $g_0$ is the surface gravitational acceleration. Eq. A2 can then be rewritten as,

$$\Delta p_i = -R_e^2 m g_0 \int\limits_{z_i}^{z_{i+1}} \frac{a_i}{(R_e + z')^2} + \frac{b_i z}{(R_e + z')^2} dz', \tag{A5}$$

with,

$$a_i = n_i - z_i \frac{n_{i+1} - n_i}{z_{i+1} - z_i}, \tag{A6}$$

and,

$$b_i = \frac{n_{i+1} - n_i}{z_{i+1} - z_i}. \tag{A7}$$

Both integrals can be computed analytically, and the delta pressures between boundaries is a linear function of number density.

The absolute pressures at each layer are also a linear function of number density,

$$\boldsymbol{p} = p_0 + \mathbf{W}\boldsymbol{n}, \tag{A8}$$

where $p_0$ is pressure at the reference altitude $z_0$, and $\mathbf{W}$ depends only on the Earth radius. Temperature at each grid point can then be evaluated with the ideal gas law,

$$T_i = \frac{p_i}{n_i k}. \tag{A9}$$

The reference pressure is not directly taken from ancillary data, instead it is evaluated using Eq. A9 using the retrieved number density and the ancillary temperature. The distinction is subtle, but we have found using reference temperature rather than pressure allows for errors in retrieved number density to more efficiently cancel.

## Appendix B: Linear error analysis

The retrieval processor provides an error estimate for the logarithm of number density, $\hat{\mathbf{S}}_{\log n}$, which is calculated through,

$$\hat{\mathbf{S}}_{\log n} = [(\mathbf{K}^T \mathbf{S}_y^{-1} \mathbf{K} + \boldsymbol{\Gamma}^T \boldsymbol{\Gamma})^{-1} \mathbf{K}^T] \mathbf{S}_y [(\mathbf{K}^T \mathbf{S}_y^{-1} \mathbf{K} + \boldsymbol{\Gamma}^T \boldsymbol{\Gamma})^{-1} \mathbf{K}^T]^{-1}, \tag{B1}$$

and can be propagated to number density as,

$$(\hat{\mathbf{S}}_n)_{ij} = n_i n_j (\hat{\mathbf{S}}_{\log n})_{ij}. \tag{B2}$$

Using Eqs. A8 and A9, we can calculate,

$$\frac{\partial T_i}{\partial n_j} = \frac{1}{n_i k} W_{ij} - \frac{T_i}{n_i} \delta_{ij}. \tag{B3}$$

The final covariance estimate is then,

$$\hat{\mathbf{S}}_T = \left[\frac{\partial T}{\partial n}\right] \hat{\mathbf{S}}_n \left[\frac{\partial T}{\partial n}\right]^T. \tag{B4}$$



*Author contributions.* DZ developed and implemented the retrieval technique. TW handled the OSIRIS processing and KD performed the intercomparisons. The first version of the manuscript was written by DZ and KD. The project was supervised by AB, DD, and ST. All authors reviewed the manuscript and contributed to discussions and conclusions.

*Competing interests.* The authors declare no competing interests.

*Acknowledgements.* This work was supported by the Natural Sciences and Engineering Research Council (Canada) and the Canadian Space Agency (CSA). Odin is a Swedish-led satellite project funded jointly by Sweden (SNSB), Canada (CSA), France (CNES), and Finland (Tekes).



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
