# Peer review of "A multi-decadal time series of upper stratospheric temperature profiles from Odin-OSIRIS limb scattered spectra"

_EGUsphere, 2023_

## Referee Comment (RC1)

**Review of the manuscript egusphere-2023-2264 "A multi-decadal time series of upper stratospheric temperature profiles from Odin-OSIRIS limb scattered spectra" by Zawada et al.**

This article presents a study on the determination of the temperature of the upper stratosphere from observations of Rayleigh scattering at the limb of the Earth's atmosphere. There is little data on the temperature in this region, which is very sensitive to climate change, and the technique of measuring Rayleigh scattering at the limb is very promising as a complement to existing techniques based on observations of atmospheric radiance in the infrared or microwave spectrum. This article makes a valuable contribution to the subject with a careful analysis of performance and error budget using Odin-OSIRIS data. I recommend its publication on EGUsphere after a minor revision detailed below.

Section 2, lines 52-55: the downward measurements are taken at around 06:00 local time, also close to dawn or dusk. Could you explain why the geometric configuration is better for observations of the bright limb in the morning part of the orbit? Is it to do with the direction of pointing in relation to the plane of the orbit? This is probably linked to the solar zenith angle (SZA) at the tangent point. Information on the SZA as a function of latitude and season is lacking for a better understanding of the observation conditions.

Section 4.1
This section describes the absolute calibration effect. It is not clear to me why the absolute calibration correction has an impact on the recovered temperature. This is obtained using a comparison with SASKTRAN simulations used as a black box, but it would be interesting to know what the physical reason is. Is it related to the estimation of the multiple scattering contribution to the radiance?

Section 4.4, lines 275-279: Is it not possible to develop an algorithm to filter the radiance profiles contaminated by PMCs and keep the non-contaminated profiles in the database?

Section 5.3, Seasonal cycle: It is difficult to understand the seasonal evolution of temperature differences between satellites from Figure 11 showing the absolute temperature for each data set. It would be useful to show the seasonal cycle of these temperature differences directly.

---

## Referee Comment (RC2)

Reivew of egusphere_2023-2264, Zawada et al. 2023

General: The hits keep coming, another new product from the OSIRIS team – middle atmosphere temperature profile, albeit over a limited altitude range ~30-60 km. The analysis method is fairly complete, but needs to supply more information about the character of the absolute calibration. The authors should consider an associated pressure profile to include with the temperature profile described here. The work is worth of publication after explaining more about the absolute calibration and minor edits are made.

Specifics:

Line 21: Must include Rusch et al. (1983), https://doi.org/10.1029/GL010i004p00261

Line 43: That will be interesting to see if the comparisons are any different given that MERRA2 assimilates Aura-MLS data.

Line 102: What profile? Be more specific

Line 132: Either "remains sensitive" or "retains sensitivity"

Line 139: How often does this (negative albedo) happen?

Line 156: When and where (lat/lon) were the simulations performed?

Line 164: Was that the only difference between versions of input radiance profiles?

Line 165: Is this the 1-sigma or 2-sigma uncertainty?

Line 166: This appears to be a very important aspect and more information is needed. What is the nature of the absolute calibration? How does it vary with time, wavelength, tangent point, latitude, etc.?
How is radiometric calibration distinguished from pointing errors in deriving the absolute calibration?

For temporal trending of the radiometric calibration have you looked at the top 1% for the brightest scenes, which would probably be associated with deep convective clouds.

Line 198: "…decreasing exponentially with decreasing altitude."

Line 199: "decrease of error with decreasing altitude…"

Figure 4: What time period was used here, 1 month, 1 season, 1 year, 10 years?

Line 207: "…the temperature error…"

Line 213: Do you mean Appendix B?

Line 224: Ok, then how does the retrieved temperature vary with wavelength?  Can the random component be reduced with multiple wavelengths?

Line 233: Any idea why this is the case? Does the retrieved number density profile behave the same way?

Line 244: How was this applied? shift entire profile by same amount for each radiance profile?

Errors in tangent height from tangent point to tangent point within a single profile will also impact the retrieved temperature since the scale height within a layer of the density profile will be wrong.

Line 249: The "precision" appears to be the absolute value of the mean difference in the left hand panel. Does that imply the precision in the original radiance profiles are 100 m or does this relationship scale with the size of the tangent point shift (is it a 1K precision for a 300 m shift)?

Section 5.1: Data from SABER might offer closer coincidences and minimize the effects of tides, since it has a precessing orbit and several times a year will sample at the OSIRIS measurement time.

Line 283: What version of MLS, UARS or Aura?

Line 288: At one point in time using the GPH was discouraged and instead using something like MERRA2 to relate altitude and pressure.

Since you retrieve a number density profile and assume a boundary condition, have you looked at a derived pressure profile to compare directly with MLS Temperature product?

Section 5.2: What time period was used for the comparisons?

Figure 10 caption: need to explain what "corr." means. Maybe "corr. = application of estimated diurnal correction.

Line 319: I am not sure Fig. 10 supports the assertion of diurnal sampling being a significant factor for the MLS and OSIRIS comparisons.  Specifically, the 'correction' increases the difference markedly for the tropics and somewhat for the Southern Hemisphere.

Line 326: Is there a sampling issue with OSIRIS favoring the summer hemisphere?

Line 328: What is the degree of correlation between the three sets?

Figure 13: What is the time period for this trend fit?

Line 341: What latitudes were used for the absolute calibration?

---

## Author Comment (AC1)

**Response to Anonymous Referee 1**

General:

This article presents a study on the determination of the temperature of the upper stratosphere from observations of Rayleigh scattering at the limb of the Earth's atmosphere. There is little data on the temperature in this region, which is very sensitive to climate change, and the technique of measuring Rayleigh scattering at the limb is very promising as a complement to existing techniques based on observations of atmospheric radiance in the infrared or microwave spectrum. This article makes a valuable contribution to the subject with a careful analysis of performance and error budget using Odin-OSIRIS data. I recommend its publication on EGUsphere after a minor revision detailed below

- We thank the reviewer for their helpful comments, following are the reviewers comments and our replies.

*Section 2, lines 52-55: the downward measurements are taken at around 06:00 local time, also close to dawn or dusk. Could you explain why the geometric configuration is better for observations of the bright limb in the morning part of the orbit? Is it to do with the direction of pointing in relation to the plane of the orbit? This is probably linked to the solar zenith angle (SZA) at the tangent point. Information on the SZA as a function of latitude and season is lacking for a better understanding of the observation conditions.*

- Sorry for the confusion here, the primary reason that only the morning measurements are used is due to the drift of the Odin-OSIRIS orbit. The orbit has generally drifted later in time with some oscillations since launch. This causes the ascending node (dusk) measurements to have very inconsistent sampling during the mission. The measurements are not of poorer quality, the sampling is just non-uniform. We have added some clarifications to the text.

*Section 4.1: This section describes the absolute calibration effect. It is not clear to me why the absolute calibration correction has an impact on the recovered temperature. This is obtained using a comparison with SASKTRAN simulations used as a black box, but it would be interesting to know what the physical reason is. Is it related to the estimation of the multiple scattering contribution to the radiance?*

- The multiple scattering estimation is definitely one effect, as the estimation of surface albedo relies on the relative calibration between the two channels, but the general effect is non-linearity of the radiative transfer equation and occurs even in single scatter. The conversion from Rayleigh scattering number density to temperature only depends on the shape of the profile, doubling the number density will have no effect on the retrieved

temperature profile. However, doubling the number density will only scale the observed radiance (by a constant factor) if the fraction of multiple scatter is constant in altitude and the optical depth along each line of sight is small so that we are in the linear regime. If either of these two conditions is not satisfied then the absolute calibration can affect the retrieved temperature. We have added some additional explanation to the text.

*Section 4.4, lines 275-279: Is it not possible to develop an algorithm to filter the radiance profiles contaminated by PMCs and keep the non-contaminated profiles in the database?*

- In theory, yes. The main problem is that it is not straightforward to develop a PMC screening algorithm for OSIRIS since the top altitude of each scan is ~65km, well below the bottom of the PMC. It is further complicated by the fact that OSIRIS takes ~90s to perform a full limb scan, if the PMC is not homogeneous in the along orbital track dimension then certain lines of sight may hit it, miss it completely, or maybe even look under it but not through it. All that said we are planning on revisiting this problem for a future data version.

*Section 5.3, Seasonal cycle: It is difficult to understand the seasonal evolution of temperature differences between satellites from Figure 11 showing the absolute temperature for each data set. It would be useful to show the seasonal cycle of these temperature differences directly.*

- The figure has been updated to show the differences. We have also indicated areas where OSIRIS does not uniformly sample the latitude band on the updated figure.

---

## Author Comment (AC2)

**Response to Anonymous Referee 2**

General:

*The hits keep coming, another new product from the OSIRIS team – middle atmosphere temperature profile, albeit over a limited altitude range ~30-60 km. The analysis method is fairly complete, but needs to supply more information about the character of the absolute calibration. The authors should consider an associated pressure profile to include with the temperature profile described here. The work is worth of publication after explaining more about the absolute calibration and minor edits are made.*

- We thank the reviewer for their helpful suggestions and their positive comments. We have added more information on the absolute calibration including a new figure. See below for more information and the responses to specific comments.

Specifics:

*Line 21: Must include Rusch et al. (1983), https://doi.org/10.1029/GL010i004p00261*

- Thank you, this has been added.

*Line 43: That will be interesting to see if the comparisons are any different given that MERRA2 assimilates Aura-MLS data.*

- We agree, we are seeing lots of interesting things, hopefully we will get something finalized soon.

*Line 102: What profile? Be more specific*

- Updated to read "number density profile"

*Line 132: Either "remains sensitive" or "retains sensitivity"*

- Fixed

*Line 139: How often does this (negative albedo) happen?*

- This happens in approximately 7.1% of scans, we have added this info to the text

*Line 156: When and where (lat/lon) were the simulations performed?*

- This is discussed in detail in a response below, but the short answer is that the specific geometry required to do this restricts what latitudes it can be done at.

*Line 164: Was that the only difference between versions of input radiance profiles?*

- Yes, we have changed the wording to make this clear

*Line 165: Is this the 1-sigma or 2-sigma uncertainty?*

- 2-sigma, we had it in the figure but not in the text, it has been added to the text.

*Line 166: This appears to be a very important aspect and more information is needed. What is the nature of the absolute calibration? How does it vary with time, wavelength, tangent point, latitude, etc.?*

- Physically we only expect the absolute calibration to vary with time and wavelength due to instrument degradations, and so we only apply a correction in time and wavelength. Because of how specific the geometry is and changes in OSIRIS operating modes/sampling/drifting local time the exact latitudes that go into the calibration are different every year. What we have done is calculate a mean calibration factor for every year, and then fit a curve through that, we have added a figure to the manuscript that shows the applied correction for the two relevant wavelengths, and some explanation about how the actual correction is calculated and applied. We realize this is not perfect, and that there is still a fair amount of uncertainty in the absolute calibration, but we believe it to be the best we can do with the data that we have. In addition we believe the elimination of significant drifts at low altitudes is good evidence that this is probably working quite well.
- That being said, we do have one year in the OSIRIS record (2004) where it seems like the absolute calibration geometry occurs fairly uniformly in latitude. As an internal check we took a look at the latitude dependence, and while there is some dependence on latitude it

is on the order of 1--2%.

[Figure]

*How is radiometric calibration distinguished from pointing errors in deriving the absolute calibration?*

- As you correctly point out, any error in the pointing would alias into a radiometric error. Fortunately the RSAS pointing correction is almost insensitive to the absolute calibration, so we don't expect it to be too large of an effect. We have added some statements to the text

*For temporal trending of the radiometric calibration have you looked at the top 1% for the brightest scenes, which would probably be associated with deep convective clouds.*

- We are a little confused about what is being suggested here. For the calibration calculation we only use high-altitudes (this information was missing from the text, it has been added) since uncertainties are too high in aerosol and ozone concentration. There may be some value in looking at how the top 1% brightest scenes have changed over time, but unfortunately is challenging for OSIRIS since the auto-exposure algorithm used has some known bugs. Frequently the brightest scenes at low-altitudes cause saturation of the detector.

*Line 198: "…decreasing exponentially with decreasing altitude."*

- Thank you, adjusted.

*Line 199: "decrease of error with decreasing altitude…"*

- Thank you, adjusted

*Figure 4: What time period was used here, 1 month, 1 season, 1 year, 10 years?*

- This is data across the entire mission, we have added this information to the caption.

*Line 207: "...the temperature error..."*

- Added

*Line 213: Do you mean Appendix B?*

- Yes thank you, fixed.

*Line 224: Ok, then how does the retrieved temperature vary with wavelength? Can the random component be reduced with multiple wavelengths?*

- The short answer is yes, that could definitely be done. Since the random component of the uncertainty on a single profile is already less than our estimated biases we did not see too much value in doing this. The ~350 nm wavelength is primarily chosen because it is used in the ozone retrieval, and we have spent a lot of time characterizing this exact pixel. We have added some extra information in the text here.

*Line 233: Any idea why this is the case? Does the retrieved number density profile behave the same way?*

- Yes, basically the temperature profile is insensitive to constant scaling of the number density, it only depends on it's shape. However a constant scaling of number density only results in a constant scaling of the observed radiance under two conditions: the fraction of multiple scatter is constant in altitude, the atmosphere is optically thin along each line of sight resulting in the radiative transfer being approximately linear. Both of these conditions start to break down as we move lower and lower into the atmosphere. Some additional explanation has been added to the text

*Line 244: How was this applied? shift entire profile by same amount for each radiance profile? Errors in tangent height from tangent point to tangent point within a single profile will also impact the retrieved temperature since the scale height within a layer of the density profile will be wrong.*

- The shift was applied essentially as a constant angular shift in the OSIRIS pointing frame, since the physical reason for the pointing uncertainty is a thermal expansion/contraction somewhere on the satellite platform. This does result in a slight change in tangent height shift from point to point, although it is relatively minor. We have no reason to suspect that the angular pointing uncertainty changes by any significant amount during the course of a single scan. We have added clarification to the text on how the pointing shift was applied

*Line 249: The "precision" appears to be the absolute value of the mean difference in the left hand panel.*

- For the pointing correction rather than simulate all potential shifts with a 1sigma of ~100m, we only simulated shifts of 100 m, which makes the precision the absolute value. This would only be true if the error is linear, which it approximately is for the pointing shift. We don't have a good quantitative estimate for our pointing uncertainty on a scan by scan basis, 100 m is only our best guess here.

*Does that imply the precision in the original radiance profiles are 100 m or does this relationship scale with the size of the tangent point shift (is it a 1K precision for a 300 m shift)?*

- Yes, the error due to the pointing shift is going to scale approximately linearly. We don't think it is likely that we do have shifts much larger than 100 m however.

*Section 5.1: Data from SABER might offer closer coincidences and minimize the effects of tides, since it has a precessing orbit and several times a year will sample at the OSIRIS measurement time.*

- Yes, very true. We are doing some additional comparisons with SABER and reanalysis for a follow on publication.

*Line 283: What version of MLS, UARS or Aura?*

- Aura, added clarification to the text

*Line 288: At one point in time using the GPH was discouraged and instead using something like MERRA2 to relate altitude and pressure.*

- Yes, this is one of the quirks of comparing with MLS. If we use MERRA2 to convert to altitude then we have to use reanalysis temperature and it is not really an independent comparison. MLS GPH has had problems in the past, constant shifts that vary in season and location. Because of this MLS v5 GPH pins the 100 hPa surface to the input reanalysis (GEOS FP-IT) since the reanalysis 100 hPa surface is likely more correct than what MLS infers. We have added some explanation on the improvements made in MLS v5

*Since you retrieve a number density profile and assume a boundary condition, have you looked at a derived pressure profile to compare directly with MLS Temperature product?*

- Yes, in the files we do include what we call "delta pressure" data product, which is essentially the estimated change in pressure from one altitude to the pinning altitude (~65 km). It is a good idea to maybe see if the comparisons with MLS can be improved by trying to use this directly, but we think it is beyond the scope of this specific paper.

*Section 5.2: What time period was used for the comparisons?*

- The overlap period between the instruments 2005-01, to 2022-12 in this case, information has been added.

*Figure 10 caption: need to explain what "corr." means. Maybe "corr. = application of estimated diurnal correction.*

- Thank you, added.

*Line 319: I am not sure Fig. 10 supports the assertion of diurnal sampling being a significant factor for the MLS and OSIRIS comparisons. Specifically, the 'correction' increases the difference markedly for the tropics and somewhat for the Southern Hemisphere.*

- We have changed the text to read "could be caused by" instead. The point of this is only to get an estimate for how big the magnitude of this effect could be. Even if the correction is making the agreement worse in some places it could still be correct, and that there are other cancelling biases.

*Line 326: Is there a sampling issue with OSIRIS favoring the summer hemisphere?*

- Maybe not an issue, but with the dawn-dusk orbit OSIRIS generally only samples the summer hemisphere. It was chosen this way to better sample the ozone hole. We have greyed out the regions on the figure where OSIRIS does not fully sample the wide latitude bin (essentially the winter hemisphere). We have adjusted the figure to indicate areas where the OSIRIS sampling is not uniform across each latitude band.

*Line 328: What is the degree of correlation between the three sets?*

- We have calculated the correlation of the anomalies between the three datasets, see below, however we find the correlation does not add much information since it is just generally higher in bins that have more natural variability, most notably areas that are strongly affected by the QBO. We have added the following statement to the text "Across all bins, the correlation between MLS and OSIRIS is rarely less than 0.7. Between ACE-

FTS and OSIRIS the correlations are weaker due to the coarse sampling, but are still greater than 0.4 in most bins."

[Figure]

*Figure 13: What is the time period for this trend fit?*

- This is the overlap period for all three instruments, so 2005-2022, information has been added to the caption.

*Line 341: What latitudes were used for the absolute calibration?*

- See the previous answers for more detail.